# Effects of Dietary Supplementation with Glutamine on the Immunity and Intestinal Barrier Gene Expression in Broiler Chickens Infected with *Salmonella* Enteritidis

**DOI:** 10.3390/ani12172168

**Published:** 2022-08-24

**Authors:** Qiujue Wu, Cong Wang, Jiahui Liao, Naizhi Hu, Binyao Cheng, Yan Ma, Yuqin Wang

**Affiliations:** College of Animal Science and Technology, Henan University of Science and Technology, Luoyang 471003, China

**Keywords:** glutamine, immune function, *Salmonella* Enteritidis, intestine barrier, broiler

## Abstract

**Simple Summary:**

*Salmonella* Enteritidis, a Gram-negative bacterium, is ingested into the gut of animals, especially young chicks. Glutamine (Gln), a nonessential amino acid, is an important energy source and active free radical-scavenging compound for enterocyte and lymphocyte cells. In this study, we found that Gln might alleviate the intestinal inflammatory response and increase the intestinal immune barrier function in broilers challenged with *Salmonella* Enteritidis. During the experimental period, supplementation with Gln improved growth performance, increased serum immunoglobulin concentrations and intestinal mucosa Bcl-2 mRNA expression levels, and decreased the lysozyme (LZM, only serum) and nitric oxide (NO) activities in serum and the intestinal mucosa, along with the intestinal mucosa barrier gene mRNA expression levels. These findings provide useful information regarding the intestinal immune barrier function of broilers.

**Abstract:**

The effects of glutamine (Gln) on immunity and intestinal barrier gene expression levels in broilers challenged with *Salmonella* Enteritidis were evaluated. A total of 400 1-day-old broilers were randomly assigned to four groups, 10 repetition treatments per group with 10 broiler chickens for a 21-day feeding trial. The groups were the normal control group (CON, no infected group, fed with a basal diet); the *S.* Enteritidis-infected control group (SCC, infected with 2.0 × 10^4^ CFU/mL of *S.* Enteritidis, fed a basal diet); and the Gln 1 and 2 groups, who were challenged with *S.* Enteritidis and fed a basal diet plus Gln at 0.5% and 1.0%, respectively. The results show that *S.* Enteritidis had adverse effects on the average daily feed intake, average daily gain, and the feed conversion ratio of infected broilers compared with those of CON broilers on d 7 (*p* < 0.05); decreased serum immunoglobulin A (IgA), immunoglobulin M (IgM), and immunoglobulin G (IgG) concentrations, and intestinal mucosa Bcl-2 mRNA expression levels (*p* < 0.05); increased the Lysozyme (LZM, only serum), NO, inducible NO synthase (iNOS) (except at 4 d), and total nitric oxide synthase (TNOS) (except at 4 d) activities in serum and the intestinal mucosa; and increased intestinal mucosa polymeric immunoglobulin receptor (pIgR) (except at 21 d), Avian beta-defensin 5 (AvBD5), AvBD14, Bax, and Bak mRNA expression levels during the experimental period (*p* < 0.05). Supplementation with Gln improved growth performance; increased serum IgA, IgG, and IgM concentrations and intestinal mucosa Bcl-2 mRNA expression levels (*p* < 0.05); decreased the LZM (only serum), NO, iNOS (except at 4 d), and TNOS (except at 4 d) activities in serum and the intestinal mucosa; and decreased intestinal mucosa pIgR (except at 21 d), AvBD5, AvBD14, Bax, and Bak mRNA expression levels during the experimental period (*p* < 0.05). These results suggest that Gln might lessen the inflammatory reaction of the small intestine and enlarge the small bowel mucosa immune and barrier function in broiler chickens challenged with *S.* Enteritidis.

## 1. Introduction

*Salmonella* Enteritidis (*S.* Enteritidis), a Gram-negative bacterium, is one of the most common and frequently isolated foodborne pathogens; it infects many animal species, including humans and poultry, resulting in enteric and systemic diseases with a high mortality rate, especially in young chicks [1]. Several observations from animal studies suggest that *S.* Enteritidis is ingested in the gut and then disrupts the composition of the intestinal microbial community. Subsequently, it interacts with the intestinal epithelial cells and deep tissues, thus damaging gut morphology and underdeveloping the gut microbiota and intestinal mucosal barrier. This results in gut permeability, immaturity of the host immune system, and depressed growth [2,3]. Moreover, in order to recruit neutrophils and maintain the intestinal mucosal barrier in response to the mucosal invasion of bacteria and toxins, epithelial cells and macrophages continuously proliferate and replace damaged intestinal epithelial cells, they express pro-inflammatory cytokines, e.g., interleukin-1β (IL-1β), and differentiate into mucus-secreting goblet cells, and finally, regulate intestine mucosa barrier genes, inflammation genes, and related apoptotic genes’ signaling pathways [3,4]. Thus, the small intestinal mucosa is considered an important binding place during infection with *Salmonella* Enteritidis.

Glutamine (Gln), a nonessential amino acid, is an important energy source and active free radical-scavenging compound for enterocyte and lymphocyte cells and is usually included in the list of “immunonutrients” [5]. Animal and clinical studies have suggested that Gln plays an important role in intestinal mucosa function and health and that it protects against stress, the invasion of pathogenic organisms, and infection and immunological challenges both in vitro [6,7] and in vivo [5,8,9]. For example, the addition of Gln can increase protein synthesis in enterocytes, promote the growth and development of the intestinal tract, regulate tight junction protein expression and intestinal immunity, rescue gut barrier function, and inhibit apoptosis or atrophy induced by stress or infection [10,11,12]. Several lines of evidence from animal studies have shown that the beneficial effects of Gln on intestinal mucosa integrity and immune system function mostly depend on plasma Gln concentration [13,14]. This is because Gln supplementation contributes to the activation of mast cells; the production of adrenal cortisol and corticotrophin-releasing factor; the release of some mediators, such as histamines, cytokines, proteoglycans, and proteases; and the amelioration of mucosal barrier dysfunction [15,16,17]. Moreover, the anti-apoptotic effects induced by the mitochondrial- or death receptor-mediated apoptotic pathway [18], the attenuated synthesis of nitric oxide (NO) by inducible NO synthase (NOS) [17], and the autophagic effect activated by some transcription factors of Gln [19] have also been found to promote intestinal epithelial cell survival during basal and physiological stress conditions [20]. Although the beneficial effects of Gln on intestinal mucosa integrity and barrier function have been documented in recent years, further studies on the anti-apoptotic effects with immunity and intestinal barrier gene expression in response to stresses in vivo are required. In this study, our aim was to determine whether glutamine changes the intestinal mucosa barrier susceptibility of broilers to *Salmonella* Enteritidis infection and to also determine the associated inflammation responses.

## 2. Materials and Methods 

### 2.1. Salmonella Enteritidis, Gln

The *S.* Enteritidis serotype was supplied by the China Veterinary Culture Collection Center (CVCC 3377, Beijing, China). The selected *S.* Enteritidis was cultured in a Luria–Bertani (LB) medium, cleaned, and then deliquated to 2.0 × 10^4^ CFU/mL in a sterile physiological salt solution. The survival cell count of infected *S.* Enteritidis was verified by the colony counting.

Gln (pharmaceutical grade: 99% purity) was purchased from Henan Honda Biological Medicine Co., Ltd., Luoyang, China, and was used in basal feed.

### 2.2. Broilers, Management, Experimental Diets, and Experimental Design

A total of 400 1-day-old Arbor Acres (AA, 51.05 ± 0.08 g) broiler chickens were weighed and randomly distributed to four groups with 10 replicates in each group and 10 broiler chickens in each replicate (half male and half female) for 21 days. The four groups were set up, that is, (1) a basal diet (CON group; *n* = 100) without added Gln and no challenge; (2) basal diet without added Gln and received 2.0 × 10^4^ CFU/mL of *S.* Enteritidis suspension (SCC, 1.0 mL per bird; *n* = 100); (3) basal diet plus 0.5% Gln and fed 2.0 × 10^4^ CFU/mL of *S.* Enteritidis suspension (Gln 1, 1.0 mL per bird; *n* = 100); and (4) basal diet plus 1.0% Gln and fed 2.0 × 10^4^ CFU/mL of *S.* Enteritidis suspension(Gln 2; 1.0 mL per bird; *n* = 100). Each broiler in the SCC, Gln 1, and Gln 2 trial groups was orally administered 1.0 mL of *S.* Enteritidis suspension on d 3, while each bird in the CON group was filled with an equivalent amount of physiological saline.

Broiler chickens were raised in metal cages (150 cm × 100 cm × 60 cm) and kept under a controlled temperature that gradually decreased from 34 °C to 22 ± 1 °C until they were 21 days of age. A 23 h lighting time was used during the d 1 to 7, and an 18 h lighting time was used during the d 8 to 21. During the trial period, the birds had unlimited access to feed and drinking water. The birds were fed corn and soybean meal-based diets in crushed form with the same nutrient components, except for the Gln. The commercial Gln was mixed into the Gln1 and Gln2 group diet. The trial diets were formulated to meet or exceed the National Research Council (NRC, 2018) nutrient recommendations for broiler chickens during d 1 to 21 (Table 1). All experiments were conducted according to the guidelines of the Institutional Animal Care and Use Committee of Henan University of Science and Technology.

### 2.3. Sample Collection and Procedures

The recordings of live weight, feed intake, and feed residues were used to evaluate the average daily feed intake (ADFI) and the feed conversion ratio (FCR) on d 4, 7, 14, and 21 of the experiment. Death rate and clinical signs of broilers were observed and noted daily during the 21-day experimental period.

Before sampling, the broiler chickens were starved for 12 h. On d 4, 7, 14, and 21, 10 chicks from each treatment group (namely: 1 chick from each replicate) were randomly selected and immediately sacrificed by cervical dislocation. Blood samples (5.0 mL) were taken from the wing vein, and the samples were centrifuged for 15 min at 1800× *g* for serum collection and then stored immediately at −80 °C for future lysozyme (LZM) and immunoglobulin (Ig) assays. Then, the carcasses of the broilers were aseptically opened, and approximately 2 cm long and 10 cm long segments of the jejunum and the ileum were cut longitudinally and flushed with cold physiological buffered saline, and the mucosa of the jejunum and the ileum were taken and divided into two part samples. One part of the sample was held at −80 °C to analyze RNA quality. The other part of the sample was held on ice until centrifugation for 10 min at 1200–1500× *g*; then, the slurry was collected and held at −80 °C until the NO and NOS enzyme assays.

### 2.4. Enzyme Activity Assay and Detection of Serum Immunoglobulin Populations

The jejunum and ileum mucosae were homogenized and centrifuged. Then, the supernatant was collected for analyses of NO and NOS levels. The activities of lysozyme (LZM, NO, and NOS in the samples were assayed using commercial assay kits (Jiancheng Bioengineering Institute, Nanjing, China).

The immunoglobulin (IgA, IgM, and IgG) were estimated with ELISA (BlueGene, Shanghai, China) methods. The IgG and IgM were examined using goat anti-chicken IgG and IgM. The test limit was 0.1 g/mL. The IgA was determined by a chicken IgA enzyme-linked immunosorbent assay.

### 2.5. Intestinal Mucosa Sample RNA Extraction and qRT–PCR Analysis

The messenger RNA abundance was determined according to the method described by Wu et al. (2018) [9]. Total RNA was collected from the jejunum or the ileum samples (2 μg) using TRIzol reagent (Invitrogen Trading (Shanghai) Co., Ltd., Shanghai, China). Tissue samples were ground using a homogenizer. The quality of total RNA was tested by 1% agarose–formaldehyde gel electrophoresis. The RNA concentration and purity in the jejunum or the ileum samples were determined from OD 260/280 readings using a spectrophotometer (Gene Quant 1300/100, General Electric Company, Boston, MA, USA). Subsequently, the total RNA in the jejunum or the ileum samples were used to reverse transcribed into cDNA using a High-Capacity cDNA Reverse Transcription Kit (Applied Biosystems, Carlsbad, CA, USA). Then, a quantitative real-time (qRT) PCR was carried out on the SYBR^®^ Green RT–PCR Kit. SYBR^®^ Green Supermix was used as the qRT–PCR master mix, and each reaction was run in duplicate. A qRT–PCR with GAPDH was the internal control. Cycling conditions were listed below: at 50 °C for 2 min; at 95 °C for 20 min; and 40 cycles at 95 °C for 10 s. The primer sequences for GAPDH, polymeric immunoglobulin receptor (pIgR), Avian beta-defensin 1 (AvBD1), AvBD5, AvBD14, Bcl-2, Bax, and Bak (all primers used in the experiment were designed and synthesized by Sangon Biotech Co., Ltd., Shanghai, China) are provided in Table 2. The relative fold change from different parts of the jejunum or the ileum was calculated using the 2 − ΔΔct method, which accounts for gene-specific efficiencies and was normalized to the mean expression of the abovementioned index [21].

### 2.6. Statistical Methods

All the data in the present study were analyzed by SPSS 21.0 (SPSS Inc., Chicago, IL, USA, 2012). Differences among the groups were analyzed by one-way ANOVA using Tukey’s multiple comparison test. All the data are expressed as the mean with standard error (SE). *p* < 0.05 were considered to be statistically significant. 

## 3. Results

### 3.1. Growth Performance 

The effects of dietary Gln supplementation on the growth performance of broilers infected with *S.* Enteritidis are displayed in Table 3. Compared with the CON group, infection with *S.* Enteritidis caused significant adverse effects on average daily gain (ADG), ADFI, and FCR (*p* < 0.05) on d 7. Compared with the SCC birds, those fed a diet with supplemental Gln had significantly improved (*p* < 0.05) ADG, ADFI, and FCR on d 7; increased (*p* < 0.05) ADG on d 7; and decreased (*p* < 0.05) ADFI and FCR on d 7. Neither *S.* Enteritidis infection nor Gln supplementation had an effect on ADFI, ADG, or FCR during the experimental period (excluding d 7).

### 3.2. LZM, IgA, IgG, and IgM Concentrations in Serum

The effects of dietary Gln supplementation on LZM, IgA, IgG, and IgM concentrations in the serum of broilers infected with *S.* Enteritidis are displayed in Table 4. *S.* Enteritidis infection increased LZM activity and decreased the concentrations of IgA, IgG, and IgM in the serum of broilers compared with those in the CON group on d 4, 7, 14, and 21 (*p* < 0.05). Dietary Gln decreased LZM activity and increased the concentrations of IgA, IgG, and IgM in the serum of broilers compared with those in the *S.* Enteritidis infection group on d 4, 7, 14, and 21 (*p* < 0.05). However, there were no differences in LZM activity or in the concentrations of IgA, IgM, and IgG in the serum of broilers between the CON and Gln groups on d 4, 7, 14, and 21 (*p* > 0.05).

### 3.3. NO and NOS Levels in Serum and Intestinal Mucosa

The effects of Gln supplementation on the NO and NOS levels in the serum and intestinal mucosae of broilers infected with *S.* Enteritidis are displayed in Figure 1. Compared with the CON group, *S.* Enteritidis infection elevated the serum and the jejunal and ileal mucosae NO activity at d 4. Dietary Gln decreased NO activity in the serum and the jejunal and ileal mucosae of broilers compared with those in the *S.* Enteritidis infection group (*p* < 0.05). However, there were no differences in NO activity between the CON and Gln groups (*p* > 0.05).

The inducible NO synthase (iNOS) and total nitric oxide synthase (TNOS) activities in the serum and the jejunal and ileal mucosae on d 4 were not affected after infection, and there were no differences among the trial groups (*p* > 0.05). 

At d 7, 14, and 21, in contrast with the control group, *S.* Enteritidis infection increased the NO, iNOS, and TNOS activities in the serum and the jejunal and ileal mucosae of broilers (*p* < 0.05). Dietary Gln decreased NO, iNOS, and TNOS activities in the serum and the jejunal and ileal mucosae of broilers compared with those in the *S.* Enteritidis infection group (*p* < 0.05). However, there were no differences in NO, iNOS, and TNOS activities between the CON and Gln groups (*p* > 0.05).

### 3.4. The Intestinal Mucosa Barrier Genes’ mRNA Expression Levels

Figure 2 displays the intestine barrier genes’ mRNA expressions of broiler chickens in the CON, *S.* Enteritidis, and Gln groups. *S.* Enteritidis infection significantly upregulated the gene expressions of pIgR, AvBD5, and AvBD14 on d 4, 7, 14, and 21 compared with the control group (*p* < 0.05). The expression of pIgR, AvBD5, and AvBD14 genes in jejunum and ileum were significantly downregulated in the Gln group compared with the *S.* Enteritidis-challenged broilers *(p* < 0.05), but no significant difference in pIgR, AvBD5, and AvBD14 mRNA expression levels were observed between control and Gln groups (*p* > 0.05). Moreover, no differences in the jejunal and ileal mucosae AvBD1 gene expressions were observed among the four treatment groups on d 4, 7, and 14 (*p* > 0.05).

On d 21, broilers in the SCC group had significantly higher gene expression levels of AvBD5 and AvBD14 in the jejunal and ileal mucosae than those in the CON group (*p <* 0.05). Moreover, Gln supplementation had a significant effect on the abovementioned genes, which were downregulated by Gln supplementation (*p <* 0.05). By contrast, no differences in the AvBD1 expression levels of the jejunal and ileal mucosae were observed between the control and Gln groups (*p* > 0.05), and there were no differences in the jejunal and ileal mucosae pIgR and AvBD1mRNA expression levels among the four treatment groups (*p* > 0.05).

### 3.5. The Related Apoptotic Genes’ mRNA Expression Levels in the Intestinal Mucosa 

The effects of dietary Gln on transcription factors in innate immune-related signaling pathways and the regulation of broilers infected with *S.* Enteritidis are displayed in Figure 3. At d 4, 7, 14, and 21, compared with the broilers in the CON group, *S.* Enteritidis infection significantly reduced the mRNA expression levels of Bcl-2 in the jejunal and ileal mucosae tissue (*p <* 0.05). Gln supplementation had a significant effect on the abovementioned gene, which was upregulated by Gln supplementation (*p <* 0.05). By contrast, no differences in the Bcl-2 expression level of the jejunal and ileal mucosae were observed between the control and Gln groups (*p* > 0.05).

On d 4, 7, 14, and 21, compared with the CON group, the mRNA expression levels of Bax and Bak were strongly upregulated in the jejunal and ileal mucosae samples from the *S.* Enteritidis infection group (*p* < 0.05). Gln supplementation significantly downregulated the gene expressions of Bax and Bak compared with the SCC group (*p* < 0.05), but no significant differences in Bax and Bak mRNA expression levels were observed between the control and Gln groups (*p* > 0.05).

## 4. Discussion

### 4.1. Effect of Dietary Gln on the Growth Performance of Broilers Infected with Salmonella Enteritidis

*Salmonella* infections have significant negative effects on the production performance, intestinal flora colonization, and gut health of broiler chickens [9,22], confirmed in this study. In the present study, *Salmonella* infection decreased the production performance of chicks on Day 7 and had no effect on the production performance of broilers in other experimental periods, but it did effectively activate the immune response system to protect against infection. Moreover, these results are consistent with those of previous studies showing that broiler chickens are extremely sensitive to *S.* Enteritidis infections in the first 7 days of life [23,24], wherein *S.* Enteritidis can effectively colonize the gut and produce systemic or septicemic disease and reduce the production performance of young birds. The effects of *S.* Enteritidis on the growth performance of broiler chickens in our present study differ slightly in results from other experiments [24]. The age of chickens, strain variations, challenge doses, and animal management conditions might effectively explain the differences that *S.* Enteritidis has on the growth performance of broiler chickens [9].

However, Gln supplementation improved ADG, ADFI, and FCR and reduced the amount of *S.* Enteritidis in the challenged chickens on d 7, and some clinical symptoms (depressed, both eyes closed, head bowed, wings drooping, motionless, etc.) vanished. The body weight gain of infected broilers improved to levels similar to those of broilers under normal feeding and management conditions on d 14 or 21, indicating that Gln might exert a protective role in controlling *Salmonella* infection, which is similar to a finding in a previous study conducted by Fasina et al. (2010) [25]. Gln, a critical gut-trophic nutrient, is a major respiratory energy source for enterocytes and has been shown to have a positive effect on gain performance in weaned pigs [26], broilers [27], and red drums [28]. These studies indicated that Gln may improve their growth performance in early life under stress conditions, which could be attributed to the beneficial effect of Gln on the growth and development of the digestive organs, the digestion of nutrients, and the improvement in apparent nitrogen retention. However, this might also be because Gln can not only directly meet the physiological need for Gln of the intestinal mucosa, but it can also improve the efficiency of protein synthesis or other physiological requirements in the diets of AA chickens [17].

On the other hand, there were no differences between Gln 1 and Gln 2; these results indicated that 0.5–1.0% Gln could meet the necessary physiological needs of the small intestine for 1–21 AA chicken. This is in keeping with the findings of Murakami et al. (2007) [29], Batal and Bartell (2007) [30], Soltan (2009) [31], and Ayazi (2014) [32], who obtained similar results with broilers from 1 to 42 days under normal or abnormal physiological conditions. These authors suggested that the optimum dose of glutamine for broilers was at about 0.5-1.0%. Soltan (2009) [31] indicated that more than 0.5% (1%) of Gln in the diet can be toxic and reduce the body weight gain of broiler chickens. Thus, in our study, the growth performance and other indexes were not affected by glutamine supplementation.

### 4.2. Effect of Dietary Gln on LZM and Antibody Levels in the Serum of Broilers Infected with Salmonella Enteritidis

LZM, as a lytic protein, is an important indicator in the cell immune system in the protection against various stresses [33]. Many studies have found that exposure to a handling stressor can increase LZM activity in the plasma of animals [34,35,36]. In the current study, a similar increase was also found in the serum LZM activity of broilers, which may suggest immunodepression due to stress caused by *Salmonella* Enteritidis and may reflect antibody changes during the development of the immune response [29]. Moreover, the increases in the serum LZM enzyme activities might suggest stress due to *Salmonella* Enteritidis, possibly resulting in LZM synthesis concerning the growth performance, immune or oxidative stress responses, and biochemical parameters in broilers. In contrast, decreases in serum lysozyme activity have been observed in stressed animals [37,38], and these differential effects could be explained by differences in the degree of stress, its intensity and duration, and the type of stressors [27]. These findings suggest that the further elucidation of lysozyme activity against various types of stress is important for a better understanding of the body–stress interaction and immunity mechanism.

However, some reports have indicated that supplementing certain nutrients (such as glucan, vitamin C, and vitamin E) can modulate lysozyme activity and provide protection against various stress in various fish species [39,40,41,42]. Liu et al. (2017) [43] also indicated that glutamine supplementation inhibits the mRNA expression of lysozyme in the jejunum or ileum of ETEC-infected mice. In the current study, a similar decrease was also found in the serum LZM activity, showing that feed with Gln significantly raised nonspecific immunity levels as measured through enhanced lysozyme activities. The activation of lysozyme activities by Gln also indicates the role of activated macrophages in Gln-induced defense, and this has a vital role in the best health of broilers by forming the foundations for a body’s immune capability and, thus, the preservation against *S.* Enteritidis infection. Interestingly, this finding is different from a previous finding indicating that glutamine supplementation encourages the jejunum or ileum lysozyme gene expression of healthy mice [44]. The underlying molecular mechanism for Gln in the cell immune system needs to be further studied; this shows the complicated relationship between intestinal nutrients, pathogens or viruses, and immune capability.

Igs, also known as antibodies, are major glycoprotein molecules produced by B lymphocytes. They provide essential immunological protection against bacterial and viral infections in the body. Consequently, we assessed the results of *S.* Enteritidis infection on Igs in the serum of broilers. The results verify that chicks with *Salmonella* experienced decreased concentrations of serum IgA, IgG, and IgM. These results show that Igs are obviously included in the processing of the serum immune reaction to *Salmonella* and are important in preserving lymphocytes against pathogens and bacterial infections [45,46]. On the other hand, we also found that Gln can increase serum Ig concentrations (IgA, IgG, and IgM). In agreement, Tian et al. (2009) [47], Fan et al. (2015) [48], and Wu et al. (2016) [49] demonstrated that Gln expands IgA-positive plasma cells of rats, which can in part contribute to the depletion of peripheral T cells [50]. However, it will be meaningful to evaluate whether Gln metabolism in the intestine also impairs the Ig-positive B cells or T cells host.

### 4.3. Effect of Dietary Gln on the NO and NOS Levels in the Serum and Intestinal Mucosa of Broilers Infected with Salmonella Enteritidis

NO is a diatomic free radical molecule involved in the eradication of bacteria, parasites, and viruses, thus being an inducer or suppressor of apoptosis, an immunoregulator, or an important signaling mediator of immune and inflammatory responses via its inhibitory or apoptotic effects on cells [51]. It has been demonstrated that several stress factor treatments can induce an endogenous increase in NO, which might be essential for stress sensing, signal transduction, and the activation of an adaptive stress response [52,53,54]. In agreement, we found that *S.* Enteritidis infection increased the NO (only on Day 4), iNOS, and TNOS activities in the serum and the jejunal and ileal mucosae of broilers at d 7, 14, and 21, indicating that the NOS/NO pathway is a crucial signaling molecule involved in immune and inflammatory responses and epithelial barrier permeability to *S.* Enteritidis stress. These data may contribute to the notion that *S.* Enteritidis affects the blood flow distribution, platelet and leukocyte adhesion, immune cell reactivity, and superoxide-generating system modulation of the layers of the gastrointestinal tract [55].

Here, we showed that Gln could eliminate the NO accumulation against *S.* Enteritidis injury and reduce the iNOS and TNOS activities in the serum and the jejunal and ileal mucosae of broilers. The findings of the present study are consistent with those of a study conducted by Meng et al. (2005) [53], which indicated that the presence of arginine decreases LPS-induced NO activity, that the intracellular NO synthesis process that is inhibited by arginine is carried out by NOS, and that the protective effect of Gln on the intestinal mucosa barrier under *S.* Enteritidis infections might be mediated by NO [56]. Combined with the Gln data, these data suggest that Gln is critical in regulating *S.* Enteritidis-induced NO synthesis. As such, Gln is a major energy source for the small intestinal mucosa, an essential precursor for the intestinal synthesis of Gln, nitric oxide, polyamines, purine, and pyrimidine nucleotides, and is required for the maintenance of intestinal mucosal mass and integrity [56]. However, further enzymological work is required to establish the biochemical bases of Gln in intestinal NO synthesis.

### 4.4. Effect of Dietary Gln on the Intestine Mucosa Barrier Genes’ mRNA Expressions of Broilers Infected with Salmonella Enteritidis

pIgR, which is part of the super-Ig family, is produced by bronchial epithelial cells, as well as the epithelia of the small and large intestines, and helps to maintain mucosal barrier integrity and intestinal homeostasis by transporting polymeric IgA antibodies across intestinal epithelial cells into gut secretions [57,58,59]. As was the case with LPS stimulation, Bruno et al. (2011) [59] found that the transcriptional activity of the pIgR reporter gene was increased. Similarly, we also found that the broilers in the *S.* Enteritidis infection group had increased mRNA expression levels of pIgR in the intestinal mucosa at 4, 7, and 14 days of age, suggesting that pIgR expression may be part of the developmental program of the intestinal immune system. These data may contribute to the understanding of the production and transport of secretory IgA [59]. Chorny et al. (2010) [60] indicated that the activation of proinflammatory chemokine and cytokine signaling pathways (such as nuclear factor-κB (NF-κB) signaling) can stimulate less conversion to IgA in mucosal B cells and increase the generation and turnover of sIgA, and thus may be responsible for sustaining or enhancing epithelial barrier function, producing antimicrobial peptides, and modulating body immune responses. Thus, we deduce that the maintenance of high-level pIgR expression can be a vital molecular mechanism through which products of the gut microbiota and host cytokines favor the integrity and barrier function of the intestinal mucosa. In contrast, it has been reported that *Escherichia coli* colonization in mouse small intestines can inhibit the intestinal immune factors (such as pIgR) of mRNA expression [43]. The potential molecular mechanism for this discrepancy is not clear, needing further study; however, this suggests the complicated connections between intestinal causative agents and the intestinal immune system.

We verified that the supplementation of glutamine reduced bacterial colonization and depressed the activation of intestinal internal immune capacity (e.g., the mRNA expression of pIgR) in the broilers of the SCC group. Similarly, Wu et al. (2016) [49] found that Gln supplementation downregulated the expression of pIgR promoters and affected the expression of pIgR mRNA in a mouse model, which suggests that Gln may inhibit intestinal *S.* Enteritidis infection through intestinal innate immunity responses; this is similar to a previous conclusion stating that dietary 0.5–1.0% Gln has the most significant immunostimulatory effects on broilers injected with *S.* Enteritidis [27]. These results can be attributed entirely to the effect of Gln on the SIgA and IgA+ contents of the small intestine, which may produce through the gut microbiota and host cytokines, followed by the T cell-dependent and T cell-independent pathways [49]. Moreover, Gln can modulate the pIgR gene expression by means of its effects on the intestinal microbiota, cytokines, and cellular signal transduction processes (such as STAT, NF-κB, and MAPK) [44]. These observations are different from previous observations indicating that Gln supplementation promotes the mRNA expression of pIgR in the jejunum or ileum of *Escherichia coli*-infected mice [53]. The potential molecular mechanism for this discrepancy is not clear, needing further study; however, it may result from differences in the experimental method or differences in the infection or stress model.

AvBDs are a bunch of small peptides of cysteine-rich repeats, commonly known as gallinacins, and are now considered to have broad antimicrobial activity in the body’s immune system and the immunoreaction process against pathogens and viruses [61,62,63,64]. The results of gene expression in vivo and an antibacterial test in vitro showed that beta-defensin genes AvBD4 and AvBD9 had very obvious effects on *Salmonellosis* resistance in chickens [59,65]. Similarly, we also found that the *S.* Enteritidis infection broilers had increased the jejunal and ileal mucosae AvBD5 and AvBD14 mRNA expression levels, indicating that the AvBD5 and AvBD14 genes variation affected the *S.* Enteritidis susceptibility, coinciding with the results from other broiler chicken challenge or stress studies, where investigators infected or stressed broiler chickens with certain particularized *Salmonella* spp. [66,67,68,69]. It has been proposed that AvBD5 and AvBD14 show antibacterial activity against *S.* Enteritidis infection, which contributes to improving the understanding of the innate immune response in the small intestine. They provide new insights into the mechanisms of tolerance and autoimmunity and their importance in physiological and pathological processes. Interestingly, the findings from the present study are inconsistent with previous data regarding the expression of AvBD genes, as there was an absence of mRNA transcripts found for AvBDs in the rooster epididymis [38]. This indicates that breed specificity, organs, and the ages of birds influence the expression of the AvBD genes [38,64].

Accumulating evidence suggests that Gln can attenuate intestinal, liver, and spleen injury by modulating the production of proinflammatory cytokines and T cell-dependent and T cell-independent pathways. *S.* Enteritidis has been suggested to stimulate lymphocyte cells in order to produce a variety of proinflammatory cytokines and antigens, which mediate the inflammatory response [70,71]. Therefore, in our current experiment, the Gln supplementation of *S.* Enteritidis-challenged broilers reduced the intestine AvBD5 and AvBD14mRNA expression levels. These results indicate that dietary Gln, as the main energy source for intestinal cells, might stimulate the synthesis of nucleic acids and provide energy for the proliferation of mononuclear cells, such as T lymphocytes, thus providing a proliferative signal to alleviate intestinal mucosa barrier damage. This may occur by promoting the proliferation and inhibiting the apoptosis of intestinal mucosa cells and by decreasing the release of pro-inflammatory cytokines in *S.* Enteritidis-infected broilers.

### 4.5. Effect of Dietary Gln on the mRNA Expressions of Related Apoptotic Genes in the Small Intestine of Broilers Infected with Salmonella Enteritidis

Bcl, BAK, and BAX are the key regulators of apoptosis, with Bcl being an anti-apoptotic member and BAK and BAX being pro-apoptotic members, which are capable of regulating the life or death of a cell that plays an essential role in a variety of infections stresses; thus, changes in Bax, BAK, and Bcl-2 expressions can directly reflect the apoptotic status of the cell. It has been reported that stress or infection can change the expressions of Bax and Bcl-2 in the gastrointestinal tract [72,73]. In the present study, *S.* Enteritidis-infected broilers had decreased Bcl-2 mRNA expression levels in the jejunal and ileal mucosae, which is in accordance with the abovementioned reports [72,73]. It has been shown that the Bax and Bcl-2mRNA expression levels may govern the sensitivity of intestinal epithelial cells to apoptotic stimuli. Thus, the Bcl-2 mRNA expression level of *S.* Enteritidis-infected broilers decreased, Bax and Bak mRNA expression levels increased, and the susceptibility of these cells to apoptosis reduced.

However, the addition of Gln to feed could upregulate the Bcl-2 mRNA expression level and downregulate Bax and Bak mRNA expression levels. Similarly, Chang et al. (2002) [73] found that Gln effectively suppressed program cell death by decreasing apoptotic features and increasing anti-apoptotic molecules (Bcl-2). These findings demonstrate that Gln can promote intestinal lymphocyte proliferation and improve immune function. These data may contribute to the understanding of the Gln effect on redox reaction preservation. Moreover, these results also suggest that Gln may behave as a survival nutrient, playing a crucial role in preventing activated T cells from undergoing apoptosis by upregulating Bcl-2 expression and decreasing *S.* Enteritidis-mediated apoptosis.

## 5. Conclusions

Taken together, our results indicate that *S.* Enteritidis affects intestinal immune function by increasing the production of NO and the activity of LZM and, consequently, increasing the intestinal permeability of the host to infection, thus enhancing our understanding of the innate immune response system in the small intestine. However, dietary supplementation with Gln could regulate the host’s innate immune response, the intestinal mucosa barrier, and the apoptotic gene signaling pathways to provide protection against *Salmonella* Enteritidis infections in young chickens. Further work is required to fully validate these specific mechanisms.

## Figures and Tables

**Figure 1 animals-12-02168-f001:**
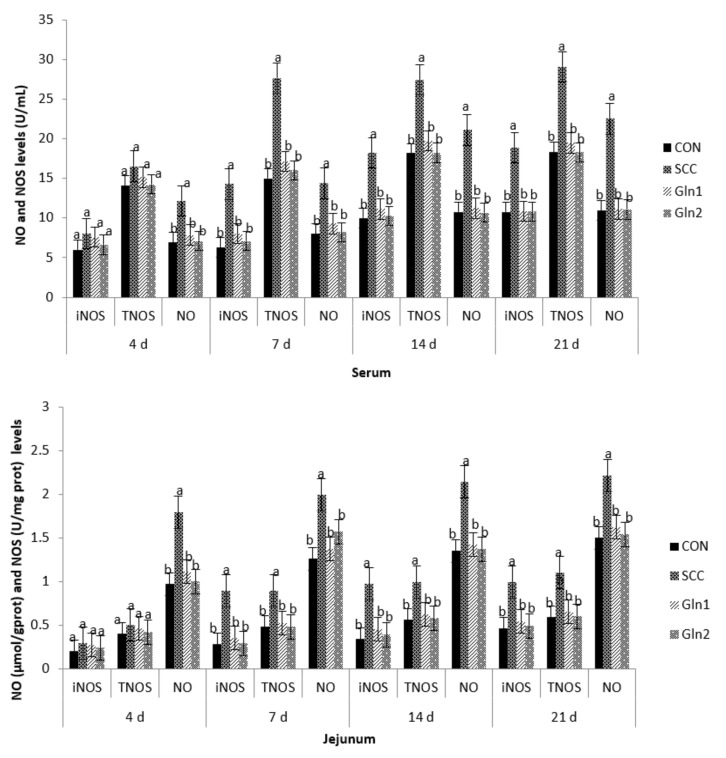
Effect of Gln on the serum and intestinal mucosa NO and NOS levels of broiler chickens that received *S*. Enteritidis. CON = a basal diet without added Gln and no challenged group, SCC = a basal diet without added Gln and received 2.0 × 10^4^ CFU/mL of *S.* Enteritidis suspension group, Gln1 = a basal diet plus 0.5% Gln and fed 2.0 × 10^4^ CFU/mL of *S.* Enteritidis suspension group, Gln2 = a basal diet plus 1.0% Gln and fed 2.0 × 10^4^ CFU/mL of *S.* Enteritidis suspension group. ^a,b^ Means within a row with different superscripts are different at *p* < 0.05. *n* = 10.

**Figure 2 animals-12-02168-f002:**
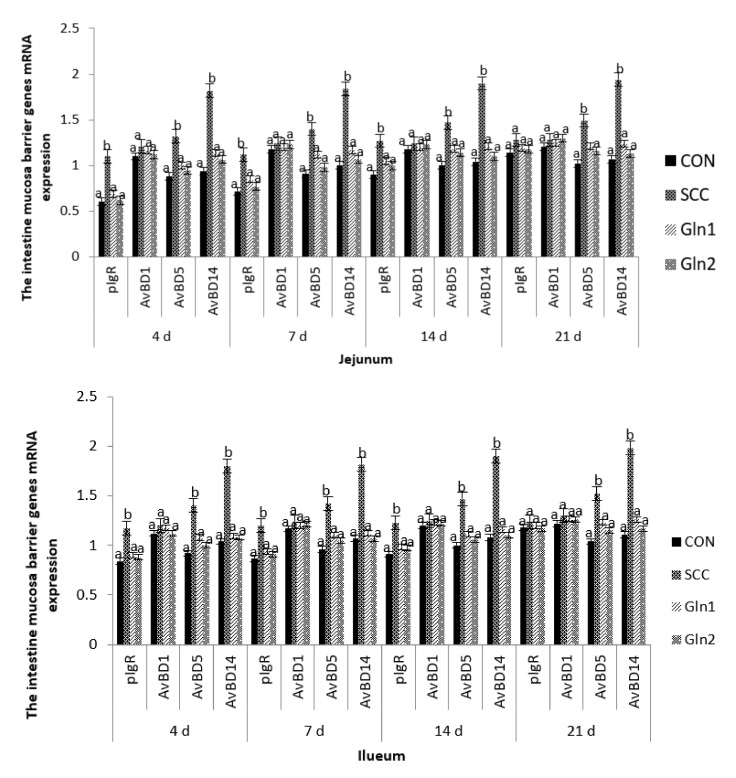
Effect of Gln on the jejunum and ileum mucosa barrier genes mRNA expression of broiler chickens that received *S*. Enteritidis. CON = a basal diet without added Gln and no challenged group, SCC = a basal diet without added Gln and received 2.0 × 10^4^ CFU/mL of *S.* Enteritidis suspension group, Gln1 = a basal diet plus 0.5% Gln and fed 2.0 × 10^4^ CFU/mL of *S.* Enteritidis suspension group, Gln2 = a basal diet plus 1.0% Gln and fed 2.0 × 10^4^ CFU/mL of *S.* Enteritidis suspension group. ^a,b^ Means within a row with different superscripts are different at *p* < 0.05. *n* = 10.

**Figure 3 animals-12-02168-f003:**
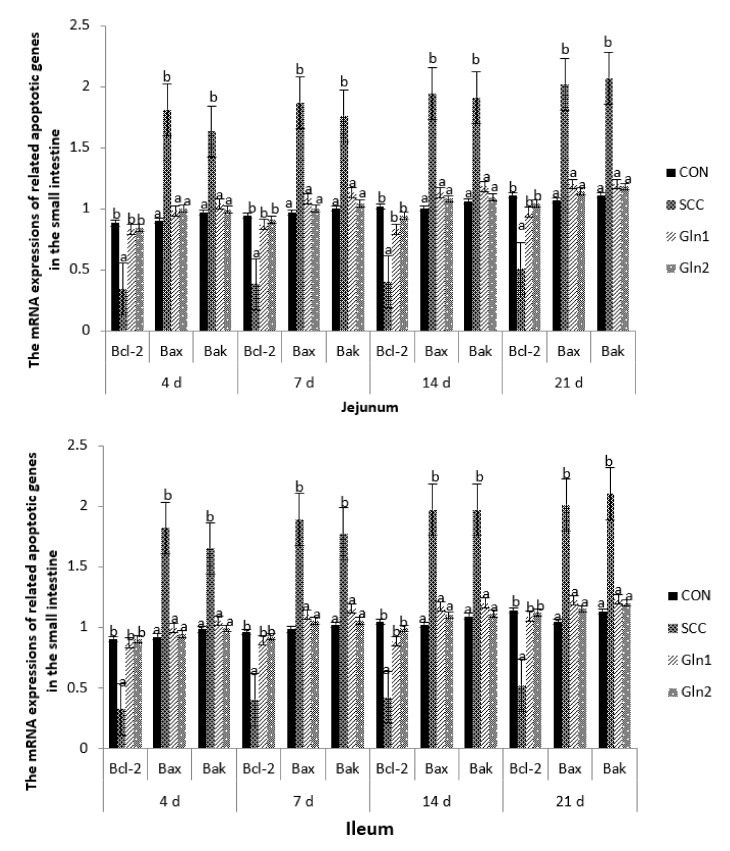
Effect of Gln on the related apoptotic genes mRNA expressions in the jejunum and ileum of broiler chickens that received *S*. Enteritidis. CON = a basal diet without added Gln and no challenged group, SCC = a basal diet without added Gln and received 2.0 × 10^4^ CFU/mL of *S.* Enteritidis suspension group, Gln1 = a basal diet plus 0.5% Gln and fed 2.0 × 10^4^ CFU/mL of *S.* Enteritidis suspension group, Gln2 = a basal diet plus 1.0% Gln and fed 2.0 × 10^4^ CFU/mL of *S.* Enteritidis suspension group. ^a,b^ Means within a row with different superscripts are different at *p* < 0.05. *n* = 10.

**Table 1 animals-12-02168-t001:** Composition of basal diets used in the trial for broilers 1–21 days of age.

Nutrient Ingredients (g/kg)	1–21 days
Corn ground	428
Soybean meal, toasted	365
Wheat	130
Soybean oil	17
Corn gluten meal	20
Canola meal	0
Na chloride	2.3
Dicalcium phosphate	15
Na bicarbonate	2.4
Ca carbonate	10.8
DL-Methionine	2.7
L-Lysine·HCl	2.2
Premix ^a^	2
Multi-enzyme	0.3
Phytase	0.3
Bentonite	0.0
Prebiotics	2
Total	1000
Calculation of nutrients (g/kg)	
Apparent metabolism energy (MJ/kg)	12.5
Crude protein	222
Calcium	9.7
Available phosphorus	4.7
Lysine	13.8
Methionine	6.0
Methionine + cysteine	8.1

Note: ^a^ Per kg: 10,000 IU vitamin A; 3000 IU Vitamin D_3_l; 30 IU vitamin E; 1.3 mg vitamin K3; 2.2 mg vitamin B1; 8 mg vitamin B2; 40 mg vitamin B3; 600 mg choline chloride; 10 mg D-pantothenate; 4 mg vitamin B6; 0.04 mg biotin; 1 mg folic acid; 0.013 mg vitamin B12; 80 mg Fe; 8 mg Cu; 110 mg Mn; 65 mg Zn; 1.1 mg iodine; 0.3 mg Se.

**Table 2 animals-12-02168-t002:** Gene sequences in real-time PCR primers.

Primer	Sequence (5′→3′)	Length
pIgR	GGATCTGGAAGCCAGCAAT	123 bp
	GAGCCAGAGCTTTGCTCAGA	
AvBD1	GGATGCACGCTGTTCTTGGT	100 bp
	TCCGCATGGTTTACGTCTGTC	
AvBD5	AGCCGATGGTATTCCTGATGG	107 bp
	TGGTGATTGTTGCCTCTGGTG	
AvBD14	TGCCGAAGATTAAGGGCAA	83 bp
	GCTAGTCCATGGTAGVAGGT	
Bcl-2	ATCCTCGCCTCCTTCGAGTT	191 bp
	ATCGCATCCTTCGTTGTCCT	
Bax	GTGCTGGCATGGGACATAGCTA	119 bp
	TGGAGTAGACCTTGCGGATAC	
Bak	ACCCGGAGATCATGGAGA	209 bp
	GATGCCTTGCTGGTAGACG	
GAPDH	GTGGTGGCCATCAATGATCC	230 bp
	ACTTGTGATCAATGGGCACG	

**Table 3 animals-12-02168-t003:** Effect of Gln on the growth performance of broiler chickens that received *S*. Enteritidis.

Item ^2^	Treatment ^1^			
	CON	SCC	Gln1	Gln2
4 d				
ADFI (g/bird/d)	20.17 ± 0.12	20.03 ± 0.23	20.54 ± 1.01	20.12 ± 0.81
ADG (g/bird/d)	15.40 ± 0.14	14.84 ± 0.14	15.33 ± 0.62	15.48 ± 0.51
FCR (g/g)	1.31 ± 0.03	1.35 ± 0.09	1.34 ± 0.10	1.30 ± 0.07
7 d				
ADFI (g/bird/d)	27.48 ± 0.16 ^a^	28.29 ± 0.13 ^b^	27.64 ± 0.14 ^a^	27.24 ± 0.08 ^a^
ADG (g/bird/d)	19.77 ± 0.14 ^b^	18.86 ± 0.10 ^a^	19.33 ± 0.05 ^b^	19.74 ± 0.13 ^b^
FCR (g/g)	1.39 ± 0.01 ^a^	1.50 ± 0.11 ^b^	1.43 ± 0.05 ^a^	1.38 ± 0.09 ^a^
14 d				
ADFI (g/bird/d)	31.71 ± 1.29	32.73 ± 0.89	32.33 ± 0.52	30.20 ± 0.91
ADG (g/bird/d)	21.57 ± 0.64	21.39 ± 0.80	21.22 ± 0.77	21.57 ± 0.74
FCR (g/g)	1.47 ± 0.09	1.53 ± 0.06	1.50 ± 0.08	1.45 ± 0.08
21 d				
ADFI (g/bird/d)	40.65 ± 0.72	38.50 ± 0.48	44.54 ± 0.81	43.52 ± 0.27
ADG (g/bird/d)	24.64 ± 0.58	21.88 ± 0.76	26.20 ± 0.47	25.90 ± 0.61
FCR (g/g)	1.65 ± 0.08	1.76 ± 0.06	1.70 ± 0.09	1.68 ± 0.03

^1^ CON = a basal diet without added Gln and no challenged group, SCC = a basal diet without added Gln and received 2.0 × 10^4^ CFU/mL of *S.* Enteritidis suspension group, Gln1 = a basal diet plus 0.5% Gln and fed 2.0 × 10^4^ CFU/mL of *S.* Enteritidis suspension group, Gln2 = a basal diet plus 1.0% Gln and fed 2.0 × 10^4^ CFU/mL of *S.* Enteritidis suspension group. ^2^ ADG, average daily gain; ADFI, average daily feed intake; FCR, feed conversion ratio. ^a,b^ Means within a row with different superscripts are different at *p* < 0.05. *n* = 10.

**Table 4 animals-12-02168-t004:** Effect of Gln on the LZM and antibodies levels in the serum of broiler chickens that received *S*. Enteritidis.

Items ^2^	Treatment ^1^			
	CON	SCC	Gln1	Gln2
4 days				
LZM (U/mL)	112.92 ± 20.42 ^a^	187.28 ± 19.87 ^b^	135.37 ± 20.86 ^a^	120.14 ± 18.24 ^a^
IgG (g/L)	3.28 ± 0.07 ^b^	1.33 ± 0.12 ^a^	3.04 ± 0.13 ^ab^	3.15 ± 0.14 ^b^
IgM (g/L)	0.63 ± 0.19 ^b^	0.36 ± 0.10 ^a^	0.60 ± 0.09 ^b^	0.63 ± 0.17 ^b^
IgA (g/L)	0.65 ± 0.16 ^b^	0.20 ± 0.15 ^a^	0.58 ± 0.14 ^b^	0.62 ± 0.14 ^b^
7 days				
LZM (U/mL)	130.31 ± 20.14 ^a^	202.86 ± 18.91 ^b^	150.32 ± 21.30 ^a^	138.34 ± 20.41 ^a^
IgG (g/L)	4.41 ± 0.10 ^b^	2.39 ± 0.13 ^a^	4.32 ± 0.09 ^b^	4.35 ± 0.11 ^b^
IgM (g/L)	0.70 ± 0.11 ^b^	0.42 ± 0.18 ^a^	0.63 ± 0.17 ^b^	0.68 ± 0.15 ^b^
IgA (g/L)	0.75 ± 0.12 ^b^	0.46 ± 0.18 ^a^	0.66 ± 0.17 ^b^	0.72 ± 0.13 ^b^
14 days				
LZM (U/mL)	142.34 ± 21.03 ^a^	214.18 ± 20.41 ^b^	168.32 ± 24.60 ^a^	147.31 ± 20.72 ^a^
IgG (g/L)	5.05 ± 0.19 ^b^	2.53 ± 0.15 ^a^	4.86 ± 0.16 ^b^	4.99 ± 0.09 ^b^
IgM (g/L)	0.84 ± 0.13 ^b^	0.48 ± 0.09 ^a^	0.75 ± 0.15 ^b^	0.80 ± 0.18 ^b^
IgA (g/L)	0.80 ± 0.15 ^b^	0.49 ± 0.10 ^a^	0.74 ± 0.14 ^b^	0.79 ± 0.11 ^b^
21 days				
LZM (U/mL)	158.63 ± 24.62 ^a^	235.85 ± 19.68 ^b^	178.37 ± 29.67 ^a^	160.31 ± 30.52 ^a^
IgG (g/L)	5.84 ± 0.20 ^b^	4.97 ± 0.11 ^a^	5.53 ± 0.18 ^b^	5.68 ± 0.17 ^b^
IgM (g/L)	0.97 ± 0.08 ^b^	0.75 ± 0.13 ^a^	0.88 ± 0.18 ^b^	0.95 ± 0.17 ^b^
IgA (g/L)	0.85 ± 0.11 ^b^	0.51 ± 0.13 ^a^	0.79 ± 0.12 ^b^	0.83 ± 0.14 ^b^

^1^ CON = a basal diet without added Gln and no challenged group, SCC = a basal diet without added Gln and received 2.0 × 10^4^ CFU/mL of *S.* Enteritidis suspension group, Gln1 = a basal diet plus 0.5% Gln and fed 2.0 × 10^4^ CFU/mL of *S.* Enteritidis suspension group, Gln2 = a basal diet plus 1.0% Gln and fed 2.0 × 10^4^ CFU/mL of *S.* Enteritidis suspension group. ^2^ LZM, lysozyme; IgG, immunoglobulin G; IgM, immunoglobulin M; IgA, immunoglobulin A. ^a,b^ Means within a row with different superscripts are different at *p* < 0.05. *n* = 10.

## Data Availability

The datasets analyzed are not publicly available due to ownership by the funding partners, but are available from the corresponding author on reasonable request.

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
