# Peer review of "Effects of Dietary Supplementation with Glutamine on the Immunity and Intestinal Barrier Gene Expression in Broiler Chickens Infected with Salmonella Enteritidis"

_animals, 2022, doi:10.3390/ani12172168_

Round 1

Reviewer 1 Report

Dear Authors,

Congratulation for the well written manuscript!

It is mentioned in lines 96-102 that the number of chickens per treatment is 60, however initially were 100. The sacrification at different ages (10/treatments) might cause the difference. Were the parameters (ADG, etc) calculated based on the replicate group means? Please state if yes. It was mentioned in l128, that 10 birds were randomly selected. From each replicate, or independently from the replicates? Were there any/no losses during the trial? In the discussion you could elaborate on the insignificance between GLN1 and GLN2. The letter size of S. Enteridis varies.

Author Response

Reviewer 1

Dear Authors,

Congratulation for the well written manuscript!

It is mentioned in lines 96-102 that the number of chickens per treatment is 60, however initially were 100. The sacrification at different ages (10/treatments) might cause the difference. Were the parameters (ADG, etc) calculated based on the replicate group means? Please state if yes. It was mentioned in l128, that 10 birds were randomly selected. From each replicate, or independently from the replicates? Were there any/no losses during the trial? In the discussion you could elaborate on the insignificance between GLN1 and GLN2. The letter size of S. Enteridis varies.

Response: dear editor, thank you very much. "n=60" has been revised into "n=100". the number of chickens per treatment should be 100, but errors were caused by careless inspection.

The design of this experiment is: 400 1-day-old chickens were selected for the experiment, which were divided into 4 treatments, 10 replicates for each treatment, and 10 chickens for each replicate. When slaughtering, one chicken is selected from each replicate for slaughtering.

During the test, no chickens died, because this dose was not a lethal dose and was determined by our long-term test. However, there were some clinical symptoms,such as: cachexia, shivering, clustering, head dropping, wing prolapse, the loss of appetite, and difficulty breathing. However, there was no death due to the short trial period and low dose.

The insignificance between GLN1 and GLN2 have been elaborated in the discussion about the growth performance.

The letter size of S. Enteridis varies have been revised.

Reviewer 2 Report

·        Line 95 please mention initial weight of chicks

·        Author mentioned that broiler chickens were randomly divided into 4 dietary treatments with 10 replicates with 10 chicks per replicate, however different The experimental treatments were mentioned (n=60) please clear this. 

·        Line 100 how dose was selected 2.0 × 104 CFU/mL of S. E?

·        Line 102 each broiler in the SCC, Gln 102 1, and Gln 2 experimental groups was orally administrated? Through drinking water? Please clear

·        Line 109-110 why lighting duration was decreased from 23 to 18 hours? Initial brooding period until 4 week of age chicks required 24 hours continuous lighting 

·        Line 130 Blood samples (5.0 mL) were taken from the caudal vein? Please confirm that birds have caudal vein or what is the correct route of blood collection in Chicken 

·        Line 161 e qRT–PCR master mix, and 161 each reaction was run in duplicate? The minimum reaction must be at least in triplicate

·        mention amount of RNA used for cDNA construction

·        Table 2 please provide BP details 

·        How primers were designed?

·        qPCR method complete detail is not provided please follow (Livak KJ,. 2001 Dec;25(4):402-8. doi: 10.1006/meth.2001.1262. )

·        Table 3 items need to revise such as ADFI, ADG, FCR, then mention abbreviations in caption 

·        Please provide a graph to show weekly average weight gain of birds in each group with different treatment 

·        It is better to convert Table 5 results in graphic bars illustration

·        Revise the mRNA results of Table 6 and 7 in graphic bars illustration separately 

·        Discussion is lengthy, revise it 

Author Response

Line 95 please mention initial weight of chicks

Response: dear editor, thank you very much. The initial weight of chicks has been added.

  • Author mentioned that broiler chickens were randomly divided into 4 dietary treatments with 10 replicates with 10 chicks per replicate, however different The experimental treatments were mentioned (n=60) please clear this. 

 Response: dear editor, thank you very much. "n=60" has been revised into "n=100". the number of chickens per treatment should be 100, but errors were caused by careless inspection.

  • Line 100 how dose was selected 2.0 × 104 CFU/mL of S. E?

Response: Thank you for your comment. The selection of the challenged dose was based on the development and experimental conditions of broilers, and some relevant literatures were referred.

These references were as followes:

(1)Wang L C , Zhang T T , Wen C , Jiang ZY,Zhou YM. Protective effects of zinc-bearing clinoptilolite on broilers challenged with Salmonella pullorum. Poult Sci, 2012, 91(8):1838-1845

(2) Wu Qiu Jue,Zheng Xiao Chuan,Wang Tian,Zhang Tie Ying. Effects of oridonin on immune cells, Th1/Th2 balance and the expression of BLys in the spleens of broiler chickens challenged with Salmonella pullorum. Research in veterinary science, 2018, 119:262-267.

(3) Qiu Jue Wu, Xiao Chuan Zheng, Tian Wang and Tie Ying Zhang*.Effect of dietary oridonin supplementation on growth performance, gut health, and immune response of broilers infected with Salmonella pullorum. Irish Veterinary Journal,2018,71:16.

  • Line 102 each broiler in the SCC, Gln 102 1, and Gln 2 experimental groups was orally administrated? Through drinking water? Please clear

Response: Thank you for your comment. We have modified the sentences in the text according to your suggestion.

 Salmonella strains were prepared into a suspension with physiological saline, and then directly orally administered with a syringe, 1 mL per chicken.

  • Line 109-110 why lighting duration was decreased from 23 to 18 hours? Initial brooding period until 4 week of age chicks required 24 hours continuous lighting 

Response: Thank you for your comment. Because the test was carried out in May, the daily sunshine time was very long, and considering the reasons of energy saving, the natural sunshine time was fully utilized during the test. 23 ~ 18h mainly refers to the artificial lighting time.

  • Line 130 Blood samples (5.0 mL) were taken from the caudal vein? Please confirm that birds have caudal vein or what is the correct route of blood collection in Chicken 

Response: Thank you for your comment. We have modified the sentences in the text according to your suggestion. Blood samples (5.0 mL) were taken from the wing vein.

  • Line 161 e qRT–PCR master mix, and 161 each reaction was run in duplicate? The minimum reaction must be at least in triplicate mention amount of RNA used for cDNA construction

 Response: Thank you for your comment. We have modified the sentences in the text according to your suggestion. PCR was performed with duplicate samples. Other datas have been added.

  • Table 2 please provide BP details 

Response: Thank you for your comment. The BP details have been added.

  • How primers were designed?
  • Response: Thank you for your comment. The details have been added. All primers used in the experiment were designed and synthesized by Sangong Bioengineering Co., Ltd.

 qPCR method complete detail is not provided please follow (Livak KJ,. 2001 Dec;25(4):402-8. doi: 10.1006/meth.2001.1262. )

Response: Thank you for your comment. The qPCR method complete detail have been added in the article. the 2ΔΔct method were only used according to the Livak KJ,. 2001 Dec;25(4):402-8. doi: 10.1006/meth.2001.1262

  • Table 3 items need to revise such as ADFI, ADG, FCR, then mention abbreviations in caption 

Response: Thank you for your comment. We have modified the word in the text according to your suggestion.

  • Please provide a graph to show weekly average weight gain of birds in each group with different treatment 

Response: Thank you for your comment. The weight gain of broilers in weekly average weight gain is mentioned in other unpublished articles. In order to avoid data repetition, different expression methods are adopted in this paper.

  • It is better to convert Table 5 results in graphic bars illustration.

Revise the mRNA results of Table 6 and 7 in graphic bars illustration separately 

Response: Thank you for your comment. Table 5, 6, and 7  have beed revised  in graphic bars illustration separately. 

  • Discussion is lengthy, revise it 

Response: Thank you for your comment. We have modified the discussion according to your suggestion.

Reviewer 3 Report

In this article, “Effects of dietary supplementation with glutamine on the immunity and intestinal barrier gene expression in broiler chickens infected with Salmonella Enteritidis (SE),” the authors wanted to study the effect of glutathione. However, in the title, glutamine is mentioned, so it is unclear whether the authors used either glutamine or glutathione. In addition, there are several issues:

 1)      Enteritidis is italicized everywhere. It is not a species name, and as it is the name of the serovar, please do not italicize.

2)     SE does not cause infection and high mortality in young chicks, so the authors must clarify their statement in lines: 45-47. The reference article mentioned in the statement does not mention high mortality or infection in young chicks. It is a flawed statement, and the authors need to provide correct evidence for the statement.

3)     The authors have mentioned glutamine in the title and the abstracts, but in the introduction, they talk about only glutathione. Did the authors use glutamine or glutathione in this study? Please explain.

4)     The title of section 2.5 (line:153) says intestinal mucosal sample for RNA extraction, but the protocol is about the only spleen, and there is no data on the spleen in the result section. Need clarification.  

5)     The authors mentioned using 400 birds for the study, but they needed only 320. Why did they use 400?

Lines 18, 31, and other places: Only acronyms are mentioned, for example: NO, iNOS. Please abbreviate them first before using acronyms.

Lines45-47; The authors have made an incorrect statement that “SE infects many animal species, including humans and poultry, resulting in enteric and systemic diseases with a high mortality rate, especially in young chicks [1]”

Author Response

Reviewer 3

In this article, “Effects of dietary supplementation with glutamine on the immunity and intestinal barrier gene expression in broiler chickens infected with Salmonella Enteritidis (SE),” the authors wanted to study the effect of glutathione. However, in the title, glutamine is mentioned, so it is unclear whether the authors used either glutamine or glutathione. In addition, there are several issues:

Response: Thank you for your comment. In our study, We used glutamine, not glutathione. Gln (pharmaceutical grade: 99% purity) was purchased from Henan Honda Biological Medicine Co., Ltd., China, and it was used in basal feed.

1)      Enteritidis is italicized everywhere. It is not a species name, and as it is the name of the serovar, please do not italicize.

Response:Thank you for your comment. We have modified the word in the text according to your suggestion.

2)     SE does not cause infection and high mortality in young chicks, so the authors must clarify their statement in lines: 45-47. The reference article mentioned in the statement does not mention high mortality or infection in young chicks. It is a flawed statement, and the authors need to provide correct evidence for the statement.

Response: Thank you for your comment. We have corrected the references according to your suggestion.

3)     The authors have mentioned glutamine in the title and the abstracts, but in the introduction, they talk about only glutathione. Did the authors use glutamine or glutathione in this study? Please explain.

Response: Thank you for your comment. In our study, We used glutamine, not glutathione. Gln (pharmaceutical grade: 99% purity) was purchased from Henan Honda Biological Medicine Co., Ltd., China, and it was used in basal feed.

I'm very sorry, due to mistakes spell, glutamine was written as glutathione. Now, we have corrected this error.

4)     The title of section 2.5 (line:153) says intestinal mucosal sample for RNA extraction, but the protocol is about the only spleen, and there is no data on the spleen in the result section. Need clarification.  

Response: Thank you for your comment. In our study, We used the jejunum and the ileum samples, not spleen. I'm very sorry, due to mistakes spell, the jejunum and the ileum samples were written as spleen. Now, we have corrected this error.

5)     The authors mentioned using 400 birds for the study, but they needed only 320. Why did they use 400?

 Response: dear editor, thank you very much. In our study, the design as follows: A total of 400 newly hatched Arbor Acres (AA ) broiler chickens were weighed and randomly divided into 4 dietary treatments with 10 replicates for a 21 d feeding trial, with 10 chicks (half male and half female) per replicate.

Due to the high mortality rate of young broilers, and ensure a certain number of samples and reduce the test error, we selected 400 chickens. However, during the whole feeding process, due to the appropriate environment, it provided better feeding and management conditions. Therefore, no chickens died during the whole experiment. The dose of Salmonella administered by gavage is only to make the chickens produce immune stress, not to make the chickens die, so only the chickens have some clinical symptoms (such as: cachexia, shivering, clustering, head dropping, wing prolapse, the loss of appetite, and difficulty breathing), but they do not die. Therefore, 400 chickens were used.

Lines 18, 31, and other places: Only acronyms are mentioned, for example: NO, iNOS. Please abbreviate them first before using acronyms.

Response:Thank you for your comment. We have added the abbreviate of these words in the text according to your suggestion.

Lines45-47; The authors have made an incorrect statement that “SE infects many animal species, including humans and poultry, resulting in enteric and systemic diseases with a high mortality rate, especially in young chicks [1]”

Response: Thank you for your comment. We have corrected the references according to your suggestion.

Reviewer 4 Report

The paper is dealing with Effects of dietary supplementation with glutamine on the immunity and intestinal barrier gene expression in broiler chickens infected with Salmonella Enteritidis. My major concerns regarding the experiment are shown below.

 Precisely determining the essential and nonessential amino acid requirements of livestock is an important task considering their significant implications on health, productivity, sustainability and environmental concerns. Any attempt contributing to this topic in the broiler chicken industry is well come.  In this context, in the present study, efforts by the authors have been directed to evaluate the implications of additive glutamine on intestinal health in broilers grown up to 21 days of age.

 In general, the study with satisfying   data, proper design and good management, microbiological and immunological observations and statements are properly supported by data presented.  The interpretations and conclusions are sound, proofed by the data and consistent with the   objectives. Briefly, the interpretations have been done with sufficient rigour to justify the conclusions,

 However, some additional information regarding the intrinsic glutamine content of practical corn-soy bean diets, as applied in the present study and the worldwide broiler industry as well, would aid the reader to better comprehend why 0.5% and 1.0% glutamine were supplemented to chicken diet. Giving the calculated glutamine content of the basal diet would be informative and beneficial while increasing the quality of the work.

Author Response

Reviewer 4

The paper is dealing with Effects of dietary supplementation with glutamine on the immunity and intestinal barrier gene expression in broiler chickens infected with Salmonella Enteritidis. My major concerns regarding the experiment are shown below.

 Precisely determining the essential and nonessential amino acid requirements of livestock is an important task considering their significant implications on health, productivity, sustainability and environmental concerns. Any attempt contributing to this topic in the broiler chicken industry is well come.  In this context, in the present study, efforts by the authors have been directed to evaluate the implications of additive glutamine on intestinal health in broilers grown up to 21 days of age.

 In general, the study with satisfying   data, proper design and good management, microbiological and immunological observations and statements are properly supported by data presented.  The interpretations and conclusions are sound, proofed by the data and consistent with the   objectives. Briefly, the interpretations have been done with sufficient rigour to justify the conclusions,

 However, some additional information regarding the intrinsic glutamine content of practical corn-soy bean diets, as applied in the present study and the worldwide broiler industry as well, would aid the reader to better comprehend why 0.5% and 1.0% glutamine were supplemented to chicken diet. Giving the calculated glutamine content of the basal diet would be informative and beneficial while increasing the quality of the work.

Response: Thank you for your comment. According to your suggestion, we added additional information about the basal diet and Gln. Thank you very much again. These datas are as follows:

The birds were provided the standard starter feed (corn–soybean meal) in mash with the same component composition, and the only difference was the Gln supplementation. Gln was supplemented and thoroughly mixed into the basal feed. The diet was formulated to meet or exceed the nutrient requirements for broilers as recommended by the National Research Council (NRC, 2018).

Round 2

Reviewer 3 Report

Looks great after the revision.